# Production Losses Associated with Alcohol-Attributable Mortality in the European Union

**DOI:** 10.3390/ijerph16193536

**Published:** 2019-09-21

**Authors:** Błażej Łyszczarz

**Affiliations:** Department of Public Health, Faculty of Health Sciences, Nicolaus Copernicus University in Toruń, 85-830 Bydgoszcz, Poland; blazej@cm.umk.pl; Tel.: +48-52-585-5409

**Keywords:** production losses, indirect costs, alcohol mortality, human capital method, European Union

## Abstract

The economic aspects of alcohol misuse are attracting increasing attention from policy makers and researchers but the evidence on the economic burden of this substance is hardly comparable internationally. This study aims to overcome this problem by estimating production losses (indirect costs) associated with alcohol-attributable mortality in 28 European Union (EU) countries in the year 2016. This study applies the prevalence-based top–down approach, societal perspective and human capital method to sex- and age-specific data on alcohol-related mortality at working age. The alcohol-attributable mortality data was taken from estimates based on the Global Burden of Disease Study 2016. Uniform data on labor and economic measures from the Eurostat database was used. The total production losses associated with alcohol-related deaths in the EU in 2016 were €32.1 billion. The per capita costs (share of costs in gross domestic product (GDP)) were €62.88 (0.215%) for the whole EU and ranged from €17.29 (0.062%) in Malta to €192.93 (0.875%) in Lithuania. On average, 81% of the losses were associated with male deaths and mortality among those aged 50–54 years generated the highest burden. Because alcohol is a major avoidable factor for mortality, public health community actions aimed at limiting this substance misuse might not only decrease the health burden but also contribute to the economic welfare of European societies.

## 1. Introduction

Alcohol is a major avoidable risk factor for mortality and morbidity worldwide [1,2,3]. As such, the misuse of alcohol is a public health concern; this is particularly relevant in the European context, because the consumption of the substance is higher there than in any other region [4,5]. Alcohol is responsible for a number of adverse health effects and is the third leading risk for the burden of disease in Europe [6]; in the European Union (EU), 1 in 7 males and 1 in 13 females aged 15–64 years die of alcohol-attributable causes, corresponding to 11.8% of all deaths in this age category [7].

This considerable health impact of alcohol translates to economic consequences which result in a decrease in the welfare of societies. The resources spent on tackling the detrimental health effects of alcohol (‘direct costs’ of health care, crime, property loss, welfare assistance, i.a.) cannot be employed for other, probably more efficient, purposes. Moreover, a certain amount of the potential economic output is not produced because of alcohol effects and this category of losses (referred to as ‘indirect costs’) encompasses premature mortality, inability to work, decreased productivity and incarceration [8,9,10,11]. Some studies also attempt to attribute economic value to pain, suffering and the decline in the quality of life due to alcohol (‘intangible costs’) [12,13]; however, this cost category does not represent either the real or potential loss of material resources.

The literature on the economic consequences of alcohol is vast and growing rapidly. Only in the last 15 years, several systematic reviews on the economic impact of alcohol consumption [4,8,10,14], heavy drinking and alcohol dependence [9,15] have been published. Studies from individual states include both estimates from high-income countries (e.g., Belgium [12,16], the United States [11,17], Germany [18,19,20], the United Kingdom [13,21], Estonia [22] and Portugal [23]) and middle-income states (e.g., Russia [24], Sri Lanka [25] and Thailand [26]). Despite this abundance of evidence, the comparability of findings from particular countries is extremely limited because of methodological heterogeneity (different definitions, data sources, cost categories and calculation methods), resulting in a broad range of estimates [8,9]. A recent systematic review of alcohol cost estimates in the EU reports an economic burden ranging from 0.12% of gross domestic product (GDP) in Portugal to 3.47% of GDP in Sweden [10]. Furthermore, two studies from the same country (Scotland) and similar period estimate the burden of alcohol at 2.03% of GDP in 2007 [27] and 1.13% of GDP in 2009–2010 [13]. Such a variation in the results makes the conclusions drawn ambiguous and precludes assessing the cost-effectiveness of alcohol-targeted policies across countries, which is one of the purposes of social cost studies [10].

This study aims to contribute to the literature by providing highly comparable estimates on one of the indirect cost categories, namely the production losses associated with alcohol-attributable mortality. Clearly, the cost of mortality constitutes only a single component of alcohol economic burden; however, several cost-of-illness studies show that its magnitude is considerable. The cost of mortality constituted the following shares of total alcohol-attributable costs (or production losses): 27.0% (or 35.2%) in Germany [19]; 30.2% (42.0%) in the US [11]; 43.8% (61.8%) in Belgium [12]; 45.0% (68.8%) in another study from Germany [18] and as much as 66.7% (69.6%) in Thailand [26]. These figures show that mortality is a major burden category in the cost of alcohol misuse. Thus, estimating the losses due to alcohol-attributable mortality in a wide range of European countries is the first contribution which provides highly comparable estimates across numerous economies. This comparability can be achieved with the benefits of data collection consistency in the EU; data regarding mortality as well as labor and economic measures is collected and reported with high uniformity. Additionally, although this study is concerned with a single component of economic burden, the indirect costs of mortality are subject to applied research in numerous health problems (particularly in cancer [28,29,30], but also in alcoholism [31]) and such estimates provide useful insights into the economic burden of illness.

Hence, the purpose of this study was to provide the cost estimates of alcohol-attributable mortality in 28 EU countries (for the year 2016) and, as such, to overcome one of the main shortcomings of the previous studies which is the incomparability of findings from various geographical settings.

## 2. Materials and Methods

Similar to a majority of previous research, this study uses the prevalence-based [18,24,26], top–down approach [27,32,33], societal perspective [34,35,36] and the human capital method (HCM) to estimate the production losses (indirect costs) associated with deaths attributable to alcohol in 28 EU countries in 2016. The top–down approach was used because for a study based on mortality data, central registries provide more accurate figures than data collected for small samples such as in the bottom–up alternative. Moreover, more previous studies on the cost of alcohol relied on the top–down approach as shown in a systematic review by Laramée et al. [9]. The research only accounts for losses in formal economy and, as such, I have not included the costs resulting from informal economic activities (e.g., housekeeping or informal caregiving) undone because of alcohol-related deaths. This approach results from the fact that no comparable data on informal production levels for the range of countries investigated here exists. Therefore, only those death cases which occurred at the working age were included in the estimates.

The choice of the HCM in cost valuation means that a premature mortality case translates to a discounted value of output that would be produced if the one who died was still alive and working until the average age of retirement [37,38,39]. A set of sex-specific, country-level labor market measures was used to identify the average time a person at a particular age would work if they had not died; these indicators were: The average age of starting first regular job (data for 2015; obtained from Eurostat on request. Data for Denmark and Sweden were not available and so, for these two countries, I used a sex-specific average value for the remaining 26 EU countries);The average effective age of exit from labor market [40]; andThe employment rate for population aged 15–64 [41].

The future values of the above measures were assumed to be constant because no reliable and comparable predictions in this respect were available for the range of countries investigated. Using these country-specific labor market statistics I identified the average time of work lost due to premature deaths associated with alcohol, separately for men and women at every 5-year interval of age for each of the 28 states.

In this study, the mean productivity of a person was proxied by per worker GDP adjusted for the decreasing marginal productivity of labor. This adjustment requires multiplying the value of production by the correction coefficient (here: 0.65) and this reflects a fact that each incremental worker produces output that is lower than on average. The use of the coefficient results from the law of diminishing marginal productivity, which—applied to the present context—means that the productivity gained thorough avoidance of early deaths would be lower than the output of an average worker in the economy (for more details see [42,43]). The GDP was also adjusted for the purchasing power parity (PPP) [44] and this allowed accounting for price differences among the 28 EU countries.

The sex-specific number of deaths at each 5-year interval from 15–19 years to 65–69 years was taken from an analysis of alcohol use and burden based on the Global Burden of Disease (GBD) Study 2016 [2]. This study estimated the number of sex- and age-specific deaths attributable to alcohol and associated with the following outcomes: atrial fibrillation and flutter; breast cancer; cirrhosis and other chronic liver diseases; colon and rectum cancer; diabetes mellitus; epilepsy; esophageal cancer; hemorrhagic stroke; hypertensive stroke disease; interpersonal violence; ischemic heart disease; ischemic stroke; larynx cancer; lip and oral cavity cancer; liver cancer; lower respiratory infections; pharynx and nasopharynx cancer; pancreatitis; self-harm; transport injuries; tuberculosis; unintentional injuries [2]. These estimates are based on the notion of population attributable fraction which represents the share of deaths that would have been avoided if the exposure to alcohol in the past had been reduced to the counterfactual level of the theoretical minimum alcohol exposure [2]. Because the number of deaths in the GBD’s estimates is reported in 5-year intervals, I assumed that each death occurred at the middle age of each interval, e.g., at the age of 47 for the 45–49 years interval. A half-cycle adjustment was applied to account for the fact that deaths occur throughout the whole year; therefore, with this assumption, the present deaths are considered to happen in the middle of the year [12,18,45].

The future costs were discounted using a 5% rate, while the potential per worker GDP growth rates for each future decade for the particular countries [40] were used to reflect the paths of their economies’ growth. No sex- or age-specific data on per worker GDP were available; thus, the estimates average the losses using data for whole populations and do not reflect the specific productivity of those dying due to alcohol misuse. This simplification might bias the results upward; this would be the case if alcohol-attributable deaths were more common among the less productive.

A one-way deterministic sensitivity analysis was performed to test how the parameter changes affect the results. A 3.5% discount rate and no discounting were used instead of the 5% in the base scenario. Moreover, I applied a ±0.05 variation in the coefficient correcting for marginal labor productivity and replaced GDP by gross value added as a measure of productivity. Additionally, the lower and upper bounds of 95% confidence intervals for the number of alcohol-attributable deaths as reported in GBD study [2] were used.

The study used only data that is publicly available or obtainable on request. It did not involve any human participants; therefore, no approval from the ethics committee was sought.

## 3. Results

### 3.1. Alcohol-Attributable Deaths at Working Age

The estimated number of alcohol-attributable deaths at working age in the year 2016 was 137,122 in the whole European Union and 79% of these cases (108,360) were deaths among men. Germany experienced the highest absolute burden of 24,137 deaths, followed by Poland (18,044) and France (16,136). On the other hand, there were less than 200 deaths at working age due to alcohol in Malta (28), Cyprus (115) and Luxembourg (120). A vast majority of these deaths concerned men and the share of their deaths in total ranged from 63.4% in the United Kingdom to 89.8% in Slovakia. The gender structure of the deaths exhibits a clear geographical and socio-economic pattern. In all the countries of Central and Eastern Europe (CEE), the share of male deaths was >84% of the total number (86.5% on average), while the corresponding proportion was 15.5; 10.5; and 8.1 percentage points lower in Northern, Southern and Western Europe, respectively. Moreover, the other (non-CEE) countries that joined the EU in the 21st century (Lithuania, Latvia, Estonia, Croatia, Slovenia and Cyprus) had an enormously high share of men’s deaths too (Table 1).

Table 1 also shows the incidence of (estimated) alcohol-attributable deaths per 10,000 population by gender; the countries have been sorted in descending order of male deaths. There were 4.34 (1.10) deaths associated with alcohol per 10,000 men (women) in the whole European Union in 2016. These deaths were relatively most common in the three Baltic states, Romania and Hungary. On the other hand, the rates for Malta were several times lower than in the countries with the greatest burden. There was not much difference in the incidence of women’s deaths between three European sub-regions (CEE, Northern and Western: 1.27; 1.24; and 1.20 per 10,000, respectively), while the rate for Southern Europe was noticeably lower (0.75 per 10,000). On the contrary, men’s mortality incidence was much more diversified between the sub-regions; the rate for the CEE countries was more than twice as large (three times) as that for the Northern and Western Europe (Southern Europe). There is also a variation in the consumption of alcohol which ranges from 7.5 liters of pure alcohol per person aged 15+ per year in Italy to 15.0 liters in Lithuania (data refers to recorded and unrecorded consumption) (Table 1).

### 3.2. Production Losses Associated with Alcohol-Attributable Deaths

The total production losses associated with alcohol-attributable mortality in 28 EU countries in 2016 were €32.1 billion adjusted for PPP (hereafter, all the Euro values are adjusted for PPP). This cost was highest in Germany (€6.5 billion), France (€4.2 billion), Poland (€3.5 billion), Romania and Spain (€2.1 billion both). On the other hand, the economic burden was lower than €100 million in Malta (€7.9 million), Cyprus (€28.0 million), Luxembourg (€74.8 million) and Slovenia (€88.9 million). The relative magnitude of the economic burden of alcohol-attributable deaths was measured by the per capita indirect cost; for the all 28 EU countries, this cost was €62.88. Lithuania was a country with the highest per capita cost of €192.93, followed by the two other Baltic states—Estonia (€143.95) and Latvia (€141.10)—as well as Luxembourg (€128.44). The per head losses were lowest in four Mediterranean countries: Malta (€17.29), Greece (€24.14), Italy (€29.45) and Cyprus (€32.84). The gender structure of the production losses clearly exhibits the gender pattern of mortality. Therefore, the costs attributable to male deaths was 81.0% of the total cost for the 28 EU countries and ranged from 67.1% (Sweden) to 91.0% (Slovakia) (Table 2).

Figure 1 shows the production losses associated with alcohol mortality in relation to GDP and this allows assessing the burden of deaths in relation to the size of the European economies. Overall, the share of GDP lost due to alcohol deaths was 0.215% for the whole EU. Four countries experienced a burden exceeding 0.5% of GDP and these were Lithuania (0.875% of GDP), Latvia (0.751%), Estonia (0.640%) and Romania (0.626%). Malta was the country with the lowest burden of 0.062% of GDP and the other countries with a cost of <0.15% of GDP were Italy (0.104%), Greece and the Netherlands (0.122% both), Sweden (0.131%) and Cyprus (0.134%) (Figure 1).

Considering the sub-regional analysis, the share of GDP lost was highest in the CEE countries (0.444%), followed by the western (0.200%), northern (0.192%) and southern states (0.146%). Noticeably, there is a high variation in this measure among the northern countries; the three Baltic states are the ones with the greatest share of GDP lost among all EU countries, while this value is much lower for other countries of the region (Figure 1).

The age distribution of losses shows that the deaths of those between 50 and 54 generated the highest burden (22.8% of the total cost among women and 20.6% among men). However, the age distribution of cost differed across the genders; for the young population (aged 20 to 34 years), the share of cost in the total losses for particular age intervals was higher for men (e.g., 5.2% for men and 3.1% for women among those aged 20–24 years). On the other hand, the corresponding shares were higher for women aged 35–54 years (Figure 2).

Figure 3 shows an association between the production losses estimated in this study and three socio-economic measures that potentially affect the magnitude of economic burden related to alcohol mortality, namely (a) per capita consumption of pure alcohol, (b) per capita GDP (€ PPP) and (c) share of GDP spent on health care. There is a substantial relationship (R^2^ = 0.393) between alcohol consumption and production losses expressed as a share of GDP. For the countries where people drink more alcohol these losses are higher; however, with growing consumption levels, the variation in costs increases. There also seems to be a relationship between the economic power of a country and the cost of alcohol deaths; the countries with higher per person GDP generally lose less in terms of the indirect costs of mortality, but this relationship is weaker (R^2^ = 0.230). Additionally, there is a negative and considerable association (R^2^ = 0.359) between expenditure on health and the production losses estimated. The countries spending a higher share of their GDP on health generally experience a lower economic burden of alcohol-related deaths and there is a higher variation in indirect cost among the countries spending lower shares of their GDP on health.

### 3.3. Sensitivity Analysis

The results of a one-way sensitivity analysis show a magnitude of changes in the present estimates resulting from the parameter variation. The use of the 3.5% discount rate resulted in an average change of 12.6% in the total production losses, with a minimum (maximum) change of 10.6% (15.5%). With no discounting, the cost estimates varied from 47.5% (Germany) to 77.1% (Cyprus), depending on the country analysis. A ±0.05 variation in the coefficient adjusting for decreasing marginal productivity changed the estimates by ±7.7%. When gross value added was used instead of GDP, the results decreased by 12.1% on average (range: −5.7% to −17.9%). The use of the lower (upper) bound of confidence interval for the estimated number of alcohol-attributable deaths reported in the GBD study [2] resulted in a change from the base scenario, ranging from −20.8% to −51.6% (20.7% to 67.9%) (Table 3).

## 4. Discussion

Using the cost-of-illness methodology, this study estimated the production losses (indirect costs) associated with alcohol-attributable mortality in 28 European Union countries in the year 2016. The results show that the health and economic burden of alcohol varies notably throughout the countries and European sub-regions.

The estimated number of deaths attributable to alcohol consumption in the EU was 137,122 in 2016 and 79% of these death cases occurred among men. The number of deaths at working age per 10,000 males was more than ten times higher in five countries—with the highest incidence (three Baltic states, Romania and Hungary) compared to Malta, where the burden was the lowest. The difference between the countries in the rate of women mortality was also high, with an eight-fold difference between the best- and worst-performing states. These alcohol-related mortality differences between the countries are clearly much deeper than for the overall mortality rates; the lowest value of 829 deaths per 100,000 inhabitants was in Spain and the respective value for the country with highest rate was <2 times greater (1602 in Bulgaria) [47]. The burden of mortality at working age was the most severe in Central and Eastern Europe and this was particularly notable for men; the lowest incidence was generally observed in Southern Europe. Moreover, a clear socio-economic pattern emerges from the present estimates; the post-communist countries that joined the EU in the 21st century experienced a much greater mortality burden and all of them had an enormously high share of male deaths. This pattern was observed in all the CEE states, the three Baltic republics as well as in Croatia and Slovenia and was previously described in epidemiological, demographic and social literature [48,49,50].

The production losses associated with alcohol-attributable deaths clearly exhibit patterns of mortality. The three countries with the greatest production losses were the Baltic republics, where the indirect costs constituted 0.640% to 0.875% of their GDP and this translated to per capita losses of €141.10–€192.93. Again, these values were several times higher than in the southern countries with the lowest costs (Malta, Greece and Italy; 0.062% to 0.122% of GDP and €17.29–€29.45 per capita). Overall, the diversity of the economic burden associated with alcohol-attributable mortality was large and this reflects not only the mortality differences but also market labor and economic conditions in particular economies. The massive mortality in the Baltic states was reinforced by relatively long labor market activity, high employment rates among women, high prospects of economic growth and more deaths at younger ages. On the other hand, the lowest costs observed in the Mediterranean countries were not only due to relatively low death rates but also because of low employment and slow dynamics of projected economic development in the future.

Considering the age distribution of mortality costs, the highest economic burden was identified for those at the age of 50–54 and 45–49 years. The age distribution of costs was similar for both genders with a slightly higher magnitude of cost shares among middle-aged women (40–54 years) and young men (see Figure 2). This gender difference clearly reflects the higher mortality of men at younger ages, which is a well-recognized phenomenon in epidemiological literature [51,52].

More generally, the higher costs associated with male mortality result from several reasons. First of all, male mortality rates are much higher than the female ones and this results from men’s higher consumption of alcohol, among others. Moreover, male employment rates are higher, and this fact strengthens the mortality effect in terms of higher economic losses among men. Possibly, also other labor market characteristics of men and women may play a role in the cost distribution by gender. Unfortunately, with the data available, I was not able to assess such issues originating from the gender differences in labor market as (potential) productivity variation between men and women. Particularly, no data on per worker GDP broken down by gender was available and the only measure that was possible to use was gender pay gap. However, I decided not to use it because it is argued that payment gap reflects rather market discrimination and not real productivity differences [53]. This choice was further justified by the fact that, in my study, productivity was proxied by GDP and not earnings.

This study also investigated the associations between indirect costs of alcohol-attributable mortality and selected socio-macroeconomic measures. According to this analysis, the countries with higher alcohol consumption suffer from greater mortality costs and this relationship is quite strong (R^2^ = 0.393). Obviously, this finding is not surprising. Moreover, the negative and notable relationship (R^2^ = 0.359) shows that the countries spending higher shares of their GDP on health experience a lower economic burden of alcohol-related mortality. This last association clearly does not prove a causal relationship; however, it suggests that investing in health care might be a way to limit economic losses of alcohol consumption. This last statement requires more insightful analysis and could be useful in formulating a hypothesis which could be tested in future research.

The stability of estimates reported here was tested using a one-way deterministic sensitivity analysis. The results were most prone to changes in the discount rate and with no discounting, the losses were 58.3% higher on average. This relatively high variation should not be surprising because a vast majority of the losses attributable to a mortality case occur in the future. Particularly, for those dying at young ages, the losses which span to decades towards the future are much lower when discounted. The other sensitivity scenario which resulted in a notable variation in results (ca. ±1/3 of costs on average) was the use of 95% confidence intervals for a number of deaths attributable to alcohol. For all other parameter changes used, the variation in estimates was not that meaningful. 

Table 4 compares the present estimates with previous findings on the costs of alcohol misuse. For this comparison, I selected those studies that referred to EU countries and reported a separate category of mortality costs. Moreover, to allow for comparability, I expressed the losses associated with alcohol-attributable mortality as a share of GDP. In some cases my estimates are higher than in the previous research (e.g., from Belgium [12,54], Ireland [55], and Portugal [56]) while for other countries, the opposite is true (e.g., Estonian [22], German [18] and French [57] estimates). It is beyond the scope of this discussion to elaborate on the sources of these differences that result from methodological variation between the studies (see, for example, the difference in discount rates in Table 4) and the year of particular studies. Nonetheless, the comparison clearly exhibits the problematic nature of drawing comparisons between the costs of alcohol in various countries. The present study aims to overcome this problem by providing estimates on the production losses associated with alcohol-related mortality for a range of countries and obtained with uniform data and methods.

## 5. Limitations of the Study

This study provides the first internationally comparable estimates on the cost of alcohol-attributable mortality for a range of countries. Despite this important contribution, the present analysis has to be interpreted with caution because some important caveats apply to both the design and the scope of the research.

Firstly, the costs estimated here only account for those alcohol-related problems that result in death. Therefore, the results do not provide figures on the overall economic burden of alcohol. Particularly, direct health care costs, crime and traffic costs as well as intangible costs, which are all a potentially important economic burden, were not considered here. Nevertheless, the assessment of mortality costs is important because they constitute a large share of total costs as explained in the introduction. Moreover, research on this single cost category is common in cost-of-illness literature [28,29,30,31].

Secondly, the number of deaths due to alcohol consumption which was taken from the GBD study was based on estimations that also have some limits of their own as pointed in the source paper [2]. These limitations were: potential failure to fully capture illicit or unrecorded production; use of American data to estimate motor vehicle harm; lack of data on harm caused to others from alcohol-attributable interpersonal violence; the exclusion of the population younger than 15 years; and inability to include some outcomes for which alcohol might be a risk factor such as dementia and psoriasis [2]. Overall, these limitations resulting from data constraints are likely to underestimate the burden of alcohol as stated in the GBD study [2].

Thirdly, there are further concerns about the appropriateness of the input data used. Particularly, the study uses average values of labor and economic measures which might bias the results. For example, the employment rate among those dying prematurely because of alcohol misuse might be different than in the total population. One might expect this rate to be lower among alcohol misusers and if this is the case, the production losses estimated here are biased upward. The same reasoning applies for the use of average per worker GDP as a productivity measure; potentially some of those dying because of alcohol might be less productive than on average. This shortcoming of my analysis has to be kept in mind when interpreting the results; however, it needs to be stressed that my approach is not different from other studies which use aggregated data. Only those studies relying on individual data can overcome this problem, but they are prone to other issues, e.g., the potentially problematic generalization of results obtained with sample data to population level.

## 6. Conclusions

This study estimated the production losses associated with alcohol-attributable deaths in 28 European Union countries in 2016. This economic burden was €32.1 billion in the whole EU and translated to the cost of €62.88 per capita. Notably, there was a great variation in both the incidence and costs of mortality due to alcohol misuse between the European countries. Because alcohol is a major avoidable factor for mortality, public health community actions aimed at limiting this substance misuse might not only decrease the health burden of alcohol but also contribute to the economic welfare of European societies which are among the heaviest drinking societies worldwide. According to the above estimates, this potential economic gain is greatest in the relatively less developed economies of Central and Eastern Europe and other post-communist states. Thus, policies aimed at decreasing alcohol consumption therein would plausibly result in more dynamic economic convergence with Western and Northern European countries.

## Figures and Tables

**Figure 1 ijerph-16-03536-f001:**
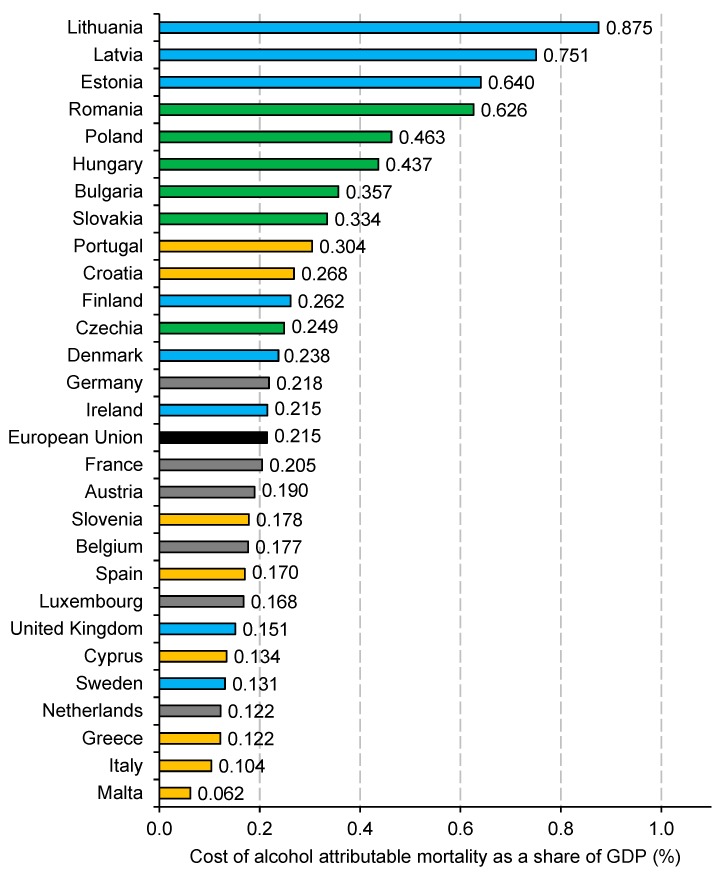
Production losses associated with alcohol-attributable mortality as a share of gross domestic product in 28 European Union countries, 2016. Notes: The bar colors mark a geographic group of countries as follows (the values in brackets show sub-regional averages): green—Central and Eastern Europe (0.444% of GDP); blue—Northern Europe (0.192% of GDP); orange—Southern Europe (0.146% of GDP); grey—Western Europe (0.200% of GDP); black—the average European Union value.

**Figure 2 ijerph-16-03536-f002:**
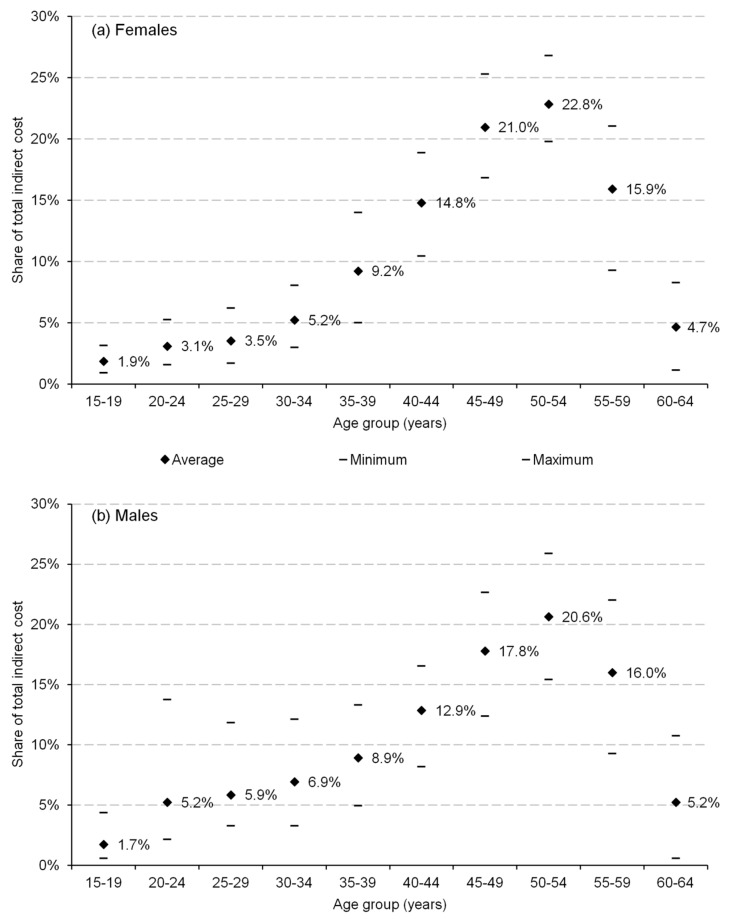
Age distribution of (**a**) female and (**b**) male production losses associated with alcohol-attributable mortality in 28 European Union countries, 2016.

**Figure 3 ijerph-16-03536-f003:**
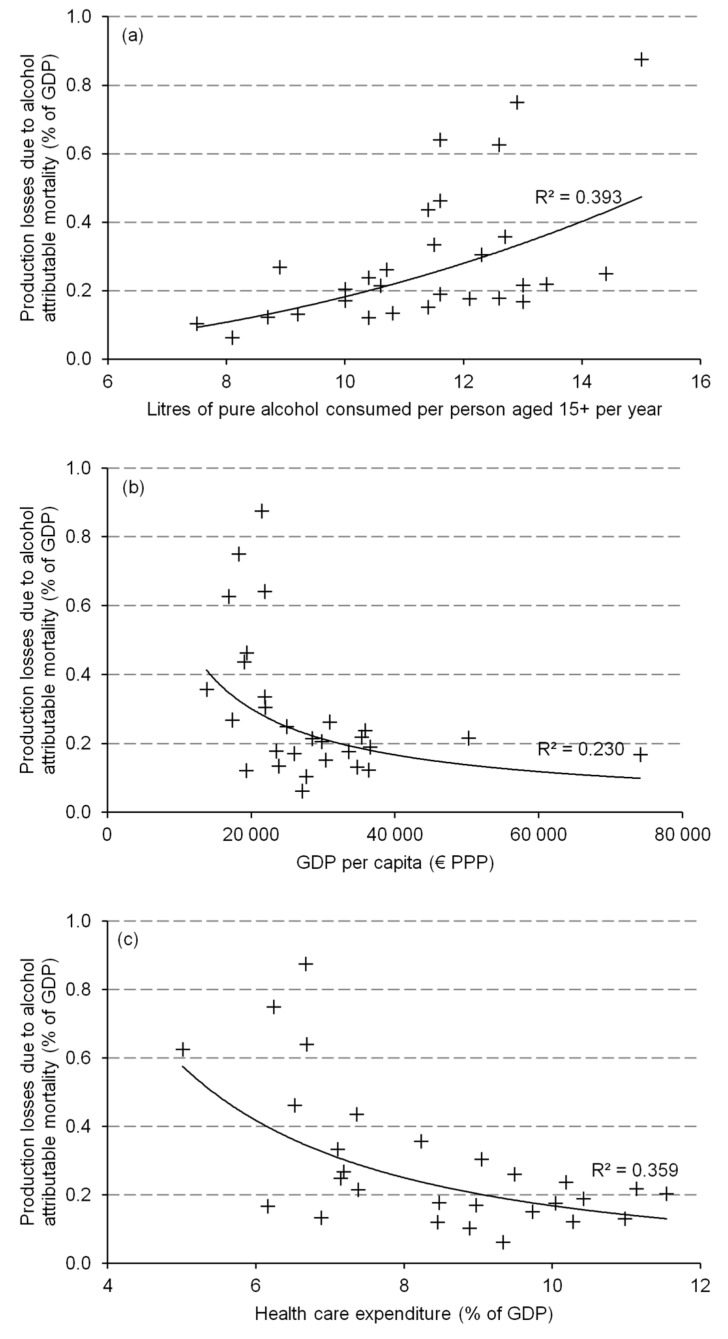
Association between production losses due to alcohol-attributable mortality and (**a**) consumption of pure alcohol per person aged 15+ per year (litres); (**b**) GDP per capita (€ PPP); and (**c**) health care expenditure as a share of GDP. Notes: GDP—gross domestic product; PPP—purchasing power parity. Source: own calculations.

**Table 1 ijerph-16-03536-t001:** Estimated number and incidence of alcohol-attributable deaths at working age and consumption of alcohol in 28 European Union countries, 2016.

	Number				Per 10,000 Population		Alcohol Consumption (Litres)
	Women	Men	Total	% of Men’s Deaths	Women	Men	
**Central and Eastern Europe**	**5923**	**38,072**	**43,996**	**86.5%**	**1.27**	**8.65**	**12.2**
Romania	1875	10,340	12,215	84.6%	1.86	10.74	12.6
Hungary	806	4734	5540	85.4%	1.57	10.11	11.4
Poland	2238	15,806	18,044	87.6%	1.14	8.60	11.6
Bulgaria	316	2727	3042	89.6%	0.86	7.87	12.7
Slovakia	214	1885	2099	89.8%	0.77	7.12	11.5
Czechia	474	2581	3055	84.5%	0.88	4.97	14.4
**Northern Europe**	**6162**	**15,086**	**21,248**	**71.0%**	**1.24**	**3.14**	**11.3**
Lithuania	337	2108	2445	86.2%	2.18	15.96	15.0
Latvia	199	1122	1321	84.9%	1.88	12.46	12.9
Estonia	144	746	890	83.8%	2.06	12.09	11.6
Finland	308	1347	1656	81.4%	1.11	4.98	10.7
Denmark	446	1265	1710	73.9%	1.55	4.44	10.4
Ireland	272	716	988	72.4%	1.13	3.04	13.0
United Kingdom	3932	6809	10,741	63.4%	1.18	2.11	11.4
Sweden	524	973	1497	65.0%	1.06	1.96	9.2
**Southern Europe**	**5243**	**19,031**	**24,275**	**78.4%**	**0.75**	**2.88**	**9.1**
Croatia	189	1323	1512	87.5%	0.88	6.57	8.9
Portugal	556	3028	3584	84.5%	1.02	6.19	12.3
Slovenia	74	413	487	84.7%	0.72	4.03	12.6
Spain	2131	7185	9316	77.1%	0.90	3.15	10.0
Greece	284	1256	1541	81.5%	0.51	2.41	10.4
Cyprus	16	99	115	85.7%	0.38	2.38	10.8
Italy	1986	5706	7692	74.2%	0.64	1.94	7.5
Malta	6	21	28	78.0%	0.27	0.94	8.1
**Western Europe**	**11,433**	**36,171**	**47,604**	**76.0%**	**1.20**	**3.95**	**11.6**
Germany	5957	18,180	24,137	75.3%	1.43	4.48	13.4
France	3578	12,558	16,136	77.8%	1.04	3.88	10.0
Austria	436	1664	2100	79.2%	0.98	3.88	11.6
Belgium	670	1835	2505	73.2%	1.17	3.29	12.1
Luxembourg	32	88	120	73.1%	1.11	3.00	13.0
Netherlands	760	1846	2606	70.8%	0.88	2.19	8.7
**European Union**	**28,762**	**108,360**	**137,122**	**79.0%**	**1.10**	**4.34**	**11.0**

Notes: Due to rounding, the ‘Total’ number of deaths might not equal the sum of values for men and women. The average numbers of deaths per 10,000 population for sub-regions and the whole EU and alcohol consumption figures (bold font used to mark that the values refer to group of countries and not single states) are population-weighted averages. ‘Alcohol consumption’ illustrates the total (recorded and unrecorded) alcohol consumption of pure alcohol consumed per person aged 15+ per year [46].

**Table 2 ijerph-16-03536-t002:** Total and per capita production losses associated with alcohol-attributable mortality in 28 European Union countries, 2016.

	Per Capita Cost (€ PPP)	Total Cost (€ PPP)			
		Women	Men	Total	% of Men’s Costs
**Central and Eastern Europe**	**87.91**	**793,907,451**	**7,172,161,352**	**7,966,068,804**	**90.0%**
Romania	108.72	216,877,078	1,925,090,280	2,141,967,358	89.9%
Poland	93.34	325,329,237	3,218,780,378	3,544,109,615	90.8%
Hungary	85.29	96,058,299	740,940,071	836,998,370	88.5%
Slovakia	75.17	36,584,184	371,650,049	408,234,233	91.0%
Czechia	63.77	83,807,460	590,058,364	673,865,824	87.6%
Bulgaria	50.63	35,251,192	325,642,211	360,893,403	90.2%
**Northern Europe**	**62.21**	**1,570,442,055**	**4,503,711,353**	**6,074,153,409**	**74.1%**
Lithuania	192.93	62,226,095	491,146,991	553,373,086	88.8%
Estonia	143.95	26,242,498	163,170,869	189,413,367	86.1%
Latvia	141.10	39,743,853	236,745,286	276,489,139	85.6%
Ireland	111.08	119,634,913	408,585,212	528,220,124	77.4%
Denmark	87.59	116,146,423	385,586,701	501,733,125	76.9%
Finland	83.25	79,100,833	378,379,100	457,479,933	82.7%
United Kingdom	47.32	975,017,362	2,128,982,695	3,104,000,057	68.6%
Sweden	46.70	152,330,079	311,114,500	463,444,579	67.1%
**Southern Europe**	**38.26**	**970,796,755**	**4,223,505,448**	**5,194,302,203**	**81.3%**
Portugal	68.73	96,220,541	613,473,616	709,694,157	86.4%
Croatia	47.91	20,113,693	179,789,308	199,903,001	89.9%
Spain	45.48	451,909,598	1,662,360,852	2,114,270,450	78.6%
Slovenia	43.06	12,082,926	76,844,881	88,927,807	86.4%
Cyprus	32.84	3,415,910	24,546,596	27,962,507	87.8%
Italy	29.45	351,154,392	1,434,396,088	1,785,550,480	80.3%
Greece	24.14	34,767,622	225,353,350	260,120,973	86.6%
Malta	17.29	1,132,072	6,740,756	7,872,828	85.6%
**Western Europe**	**68.96**	**2,779,978,519**	**10,108,405,118**	**12,888,383,637**	**78.4%**
Luxembourg	128.44	17,174,562	57,580,230	74,754,791	77.0%
Germany	79.35	1,486,429,987	5,048,038,348	6,534,468,335	77.3%
Austria	71.30	110,935,938	511,981,436	622,917,374	82.2%
France	62.69	807,345,501	3,383,954,136	4,191,299,638	80.7%
Belgium	60.76	157,787,776	530,748,450	688,536,227	77.1%
Netherlands	45.59	200,304,755	576,102,518	776,407,273	74.2%
**European Union**	**62.88**	**6,115,124,781**	**26,007,783,271**	**32,122,908,052**	**81.0%**

Notes: € PPP—euro adjusted for purchasing power parity. The average per capita costs and the share of costs attributable to men’s deaths for sub-regions and the whole EU (bold font used to mark that the values refer to group of countries and not single states) are population-weighted averages.

**Table 3 ijerph-16-03536-t003:** Sensitivity analysis for estimates of production losses associated with alcohol-attributable mortality in 28 European Union countries, 2016.

	Average Change from BS ^1^	Minimum Change from Base Scenario	Maximum Change from Base Scenario
Discount rate (BS: 5%)			
3.5%	12.6%	10.6% (Germany)	15.5% (Cyprus)
0%	58.3%	47.5% (Germany)	77.1% (Cyprus)
Coefficient to adjust for decreasing marginal labor productivity (BS: 0.65)			
0.6	−7.7%	−7.7% (all countries)	
0.7	7.7%	7.7% (all countries)	
Productivity measure (BS: gross domestic product)			
Gross value added	−12.1%	−5.7% (Ireland)	−17.9% (Croatia)
95% confidence interval for the number of alcohol-attributable deaths (BS: point estimate)			
Lower bound	−31.7%	−20.8% (Portugal)	−51.6% (Malta)
Upper bound	34.9%	20.7% (Lithuania)	67.9% (Malta)

^1^ Notes: BS—base scenario; results for BS shown in Table 2.

**Table 4 ijerph-16-03536-t004:** Comparison of production losses associated with mortality in selected studies from the European Union (2000 and onwards).

Study	Country (Year)	Losses in the Compared Study ^1^	Losses in the Present Study	DS
		% of GDP		
Lievens et al. [54]	Belgium (2012)	0.134%	0.177%	n.a.
Verhaeghe et al. [12]	Belgium (2012)	0.103%	0.177%	0%
Byrne [55]	Ireland (2007)	0.056%	0.215%	n.a.
Saar [22]	Estonia (2006)	1.825% (1.052%) ^2^	0.640%	4% (10%)
Konnopka, König [18]	Germany (2002)	0.497%	0.218%	5%
Lima, Esquerdo [56]	Portugal (1995)	0.083%	0.304%	5%
Fenoglio et al. [57]	France (1997)	0.704%	0.205%	6%
Jarl et al. [58]	Sweden (2002)	0.119% (0.331%) ^3^	0.131%	3%
Cabinet Office [59]	UK (2001)	0.200% (0,220%) ^4^	0.151%	n.a.

Notes: ^1^ Own calculations based on Eurostat’s data on GDP and costs taken from respective studies; DS—discount rate; ^2^ 4% (10%) discount rates; ^3^ net (gross) losses; ^4^ low (high) estimate; n.a.—details not available/reported.

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
