# Peer review of "Production Losses Associated with Alcohol-Attributable Mortality in the European Union"

_ijerph, 2019, doi:10.3390/ijerph16193536_

Round 1
Reviewer 1 Report
The article is well presented. The idea of article is attractive and state-of -the-art is also perfect. Therefore, I recommend to publish this article as such in the journal.

Reviewer 2 Report
Title: Production losses associated with alcohol-2 attributable mortality in the European Union
The paper is well-written, with a clear message, my only question backs to the gender differences, on page 12 line 305-306, author has stated that “This gender difference clearly reflects higher mortality of men at younger ages which is a well-recognized phenomenon in epidemiological literature”, the question is:
The gender differences in this study – specially on the losses associated with alcohol – back to lower employment rate or not equal payment to women (vs. equal age men --), please clarifies that this diff back to gender inequality treatment in the labor market or behavioral pattern. You may also like to run a sensitivity analysis and use the men rates (employment and payments or even consumption) and see the differences.
Also, add the average liters of alcohol (preferably for men and women) to T1 or T2, it helps a reader to find more about the alcohol consumptions in each country.
